# ITAS: Integrated Transcript Annotation for Small RNA

**DOI:** 10.3390/ncrna8030030

**Published:** 2022-05-02

**Authors:** Alexey Stupnikov, Vitaly Bezuglov, Ivan Skakov, Victoria Shtratnikova, J. Richard Pilsner, Alexander Suvorov, Oleg Sergeyev

**Affiliations:** 1Moscow Institute of Physics and Technology, 141701 Moscow, Russia; 2National Medical Research Center for Endocrinology, 115478 Moscow, Russia; 3Belozersky Institute of Physico-Chemical Biology, Lomonosov Moscow State University, 119992 Moscow, Russia; vitaly1530@gmail.com (V.B.); vtosha@yandex.ru (V.S.); asuvorov@schoolph.umass.edu (A.S.); 4Faculty of Bioengineering and Bioinformatics, Lomonosov Moscow State University, 119992 Moscow, Russia; vanya.skakov@yandex.ru; 5Department of Obstetrics and Gynecology, Wayne State University School of Medicine, Detroit, MI 48201, USA; rpilsner@wayne.edu; 6Department of Environmental Health Sciences, University of Massachusetts, Amherst, MA 01003, USA

**Keywords:** transcript annotation, differential gene expression, small RNA, microRNA, piRNA, tRNA-derived small RNA, RNA-seq, small RNA fragments, transcriptomics

## Abstract

Transcriptomics analysis of various small RNA (sRNA) biotypes is a new and rapidly developing field. Annotations for microRNAs, tRNAs, piRNAs and rRNAs contain information on transcript sequences and loci that is vital for downstream analyses. Several databases have been established to provide this type of data for specific RNA biotypes. However, these sources often contain data in different formats, which makes the bulk analysis of several sRNA biotypes in a single pipeline challenging. Information on some transcripts may be incomplete or conflicting with other entries. To overcome these challenges, we introduce ITAS, or Integrated Transcript Annotation for Small RNA, a filtered, corrected and integrated transcript annotation containing information on several types of small RNAs, including tRNA-derived small RNA, for several species (*Homo sapiens*, *Rattus norvegicus*, *Mus musculus*, *Drosophila melanogaster*, *Caenorhabditis elegans*). ITAS is presented in a format applicable for the vast majority of bioinformatic transcriptomics analysis, and it was tested in several case studies for human-derived data against existing alternative databases.

## 1. Introduction

### 1.1. sRNAs in Biology and Medicine

Small non-coding RNA (sRNA) are less than 200 nucleotides in length and they are important molecules in RNA silencing pathways, the regulation of gene expression and chromatin modifications [1,2,3]. Research on sRNAs has accelerated in recent decades and sRNAs have been utilized as markers of human diseases such as neurological conditions [4], cancer [5] and infertility [6,7], and in the identification of molecular biomarkers of associations between environmental exposures and health/disease outcomes [8,9]. Identification of sRNA in germ cells is of particular interest as they represent an additional source of parental hereditary information beyond DNA sequences and may have a potential role in programming offspring health [10,11]. Types of small RNA include, among others, microRNA (miRNA), piwi-interacting RNA (piRNA), small rRNA-derived RNA (rsRNA) and tRNA-derived small RNA (tsRNA) [2]. There is a wide range of techniques for small RNA detection in various somatic and germ cells, cell cultures, biofluids and extracellular vesicles, which include RT-PCR, Northern blotting, microarrays (for known sRNAs) and high-throughput next-generation sequencing, which requires downstream bioinformatics analysis [12,13].

### 1.2. Expression and Fragment Analysis of RNA-seq Data

Transcriptomics analysis of sRNA expression following RNA sequencing is a new and developing field. Several tools and pipelines analyzing specific sRNA biotypes have been designed [14,15,16,17,18], with only a few approaches that allow for the simultaneous analysis of different sRNA biotypes [19,20,21,22].

One of the key steps in the analysis consists in the recruiting of annotation data from databases that contain information on transcript loci and sequences. It has been demonstrated that the choice and alterations in the annotation structure used for expression analysis may have a significant impact on the analysis’ reproducibility [23,24]. In addition, it was argued that various transcriptomics approaches may have different performance depending on the experimental design and studied biological factors [25,26].

Recent approaches [14,15,16,17,18,22] involved the sequential mapping of reads against the transcripts of various databases. In this approach (so-called ‘map and remove’), reads mapped to transcripts in one database are no longer considered for mapping to transcripts of the next ones. This procedure results in an expression matrix, or count matrix, that may be further analyzed with differential gene expression techniques [27,28,29]. These methods recruit annotation at its sequence level, which is employed for the alignment procedure.

The ’map and remove’ technique has certain advantages (such as addressing the problem of multiple loci existing for a single transcript) as well as disadvantages (the order of databases used to sequentially align reads can affect the analysis outcome, and different sRNA biotypes are not treated independently). The latter problem could be potentially addressed by employing genome alignment (e.g., bowtie [30] and Rsubread [31]) or pseudo-alignment (such as Kallisto [32]) methods. However, the applicability of such methods to existing annotation databases has not been studied. Hence, expression analysis of small RNA demands transcript annotations that (1) provide validated and reliable information on transcripts; (2) secure the reproducibility and robustness of the analysis; and (3) allow for flexibility in the choice of bioinformatic approaches.

Thus, the objectives of the current research are to explore the actual structure and properties of existing small RNA annotation databases and to generate a common database satisfying these criteria.

## 2. Results

### 2.1. Database Summary

Several data resources have been selected as sources of annotations for specific RNA biotypes. They include miRBase [33] for microRNA, piRNAdb [34] for piRNA, GtRNAdb [35] for tRNA, UCSC database [36] for rRNA, tRFdb [37] for *Mus musculus*, *Drosophila melanogaster* and *Caenorhabditis elegans* tsRNA and MINTbase [38] for *Homo sapiens* tsRNA. All mentioned databases were accessed in September 2021.

The following issues were identified in the process of assessment of these databases.

#### 2.1.1. Annotation Entry Data Are Incomplete

Based on our analysis of existing databases, most of them contain incomplete information on certain transcripts. Table 1 and Figure 1 present statistics on missing data for human transcripts in considered databases. For certain transcripts, only loci or only fasta data are provided (for human sRNA annotation, 2318 microRNA and 187 tRNA had no sequence data, which is almost 48% and 30% of all transcripts) and this thereby limits the types of bioinformatic analysis that could be conducted.

#### 2.1.2. In-Transcript Data Conflicts

The second problem arises from the data structure provided in the annotation for a given transcript (Figure 2). MicroRNA (Figure 2A), piRNA (Figure 2B) and rRNA (Figure 2D) databases contain multiple loci corresponding to the same transcript. Transcript abundance quantification is problematic for such transcripts as they cannot be uniquely assigned to a single genomic region using genome alignment-based methods, such as featureCounts [31] or HTSeq [39]. Thus, the problem of multiple loci per transcript reduces the flexibility of bioinformatic pipelines for sRNA research.

Another identified problem is that, for many transcripts (tsRNAs in MINTbase for *Homo sapiens* and various small RNA types in other species), the length of the locus does not match the length of the corresponding fasta sequence (Figure 3A–C). This occurs due to the lack of precision in initial database loci coordinates and LiftOver transformation for genomic coordinates from hg19 to hg38 genome versions. These effects were corrected in the final ITAS version. In addition to this, 1718 transcripts in MINTbase had significant differences between sequences in database-derived fasta files and actual nucleotide sequences in corresponding genome loci (Figure 3D).

#### 2.1.3. In-Database Transcript Data Conflicts

The third problem is caused by intersecting loci of transcripts in the same database. For human annotation, piRNA entries had 388,826 loci that had intersections with other piRNAs. In such a case, reads aligned to the region shared by these transcripts cannot be unquestionably assigned to any of them. To allow for reproducible and unambiguous analysis, intersecting loci were identified and filtered out.

#### 2.1.4. Inter-Database Transcript Data Conflicts

Likewise, the fourth problem comes from the intersection of a transcript’s loci, originating from different databases. *Homo sapiens* piRNA transcripts had 1268 entries that had intersections with sRNAs of other types.

### 2.2. Integrated Transcript Annotation for Small RNA (ITAS)

The following steps were invoked to approach the outlined problems:Retrieve and fill in missing data for the incomplete entries;Approach the problem of multiple loci per transcript;Identify, correct if possible and filter out otherwise entries for transcripts with conflicting fasta-delivered data and locus-delivered data;Identify and filter out in-database loci-wise intersecting entries;Identify and filter out inter-database loci-wise intersecting entries.

The identified problems were corrected where possible and entries with severe conflicts were filtered out. The statistics for human annotation correction for different biotypes of small RNA are provided in Table 2. The statistics for *Mus musculus*, *Rattus norvegicus*, *Drosophila melanogaster*, *Caenorhabditis elegans* are provided in Appendix A, respectively.

The statistics for tsRNA annotation correction for four species, in turn, are provided in Table 3.

The Integrated Transcript Annotation for Small RNA (ITAS) contains information on loci and sequences of integrated transcripts that had no issues, or the issues were fixed.

In the process of integrating data from different sources, those with conflicting entries (such as transcripts with intersecting loci) need to be filtered out (correction events *Interbase conflict* in Table 2). However, this may remove certain information that may be of interest in studies dedicated to a particular sRNA type with no focus on others. Therefore, we present our results for every considered organism in three different forms:Integrated annotation data for different biotypes of sRNA with removed inter-database conflicts (statistics presented in Table 4);Separate annotations for specific small RNA types with no inter-database conflict filtering applied (statistics presented in Table 5);Separate annotation for tsRNA analysis (statistics presented in Table 5, last column).

To address the issue of multiple loci for the same transcript, the "exon” feature from gtf-format was used to optimize count summarization: reads mapped to the different loci of one transcript in this way can be summarized altogether.

The summary for the final ITAS is presented in Table 4 and Table 5.

### 2.3. Case Studies

To validate and test our annotation applicability and quality, we conducted case studies on three publicly available datasets of human sperm RNA-seq data [40,41,42], for which we ran differential expression analysis of sRNA transcripts and, separately, tsRNA analysis, using investigated factors in each study. To do so, we ran the SPORTS [19] pipeline with default settings and built-in annotation, genome alignment (bowtie [30] and Rsubread [31]) with ITAS for sRNAs and pseudo-alignment (Kallisto [32]) of reads mapped to tRNA only by Rsubread for tsRNA. Differential expression results for three cases and both pipelines are provided in Appendix A.

The results demonstrate the advantage of ITAS. MicroRNA are presented as precursors (hsa-mir-XXX) in SPORTS, but in ITAS, mature microRNA (hsa-miR-XXX-3p and/or hsa-miR-XXX-5p, or hsa-miR-XXX only) were included. Thus, analysis using ITAS allowed the identification of mature microRNAs, which can be used for further analysis—for example, for the search of gene targets and enrichment analysis. SPORTS tsRNA are shown as 5 or 3 ends of tRNA without further details, while every tsRNA has its own ID, locus and sequence in ITAS.

Using sRNA data from three literature sources, the differential expression was done by factors described in corresponding articles (Appendix A, Table 6). Here, results using Donkin and colleagues’ data are presented (Figure 4 and Figure 5 and Table 7). The SPORTS pipeline identified fewer transcripts, with a *p*-value < 0.05, than the ITAS-based genome alignment pipeline (Table 6). For Donkin et al.’s [40] data, microRNA transcript hsa-mir-155 was reported by both pipelines (Figure 4 and Figure 5 and Table 7). Several microRNAs (hsa-let-7b, hsa-mir-892c) were identified by both pipelines for Ingerslev et al.’s [41] data (Appendix A).

Using tsRNA data from Donkin et al. [40], the SPORTS pipeline identified more tsRNA, with adjusted *p*-value < 0.1, than the ITAS based genome alignment pipeline, but from Ingerslev et al. [41] and Hua et al.’s [42] data, the SPORTS pipeline identified less tsRNA. Moreover, only ITAS employing the genome alignment pipeline was able to find differentially expressed tsRNA for Hua et al.’s [42] data.

## 3. Materials and Methods

### 3.1. microRNA, piRNA and tRNA Processing

Files with mature transcripts sequences in fasta format (referred to as fasta sequences) and annotation of transcripts in bed format (loci) were obtained from corresponding databases: GtRNAdb Data Release 19 (June 2021) [35] for mature tRNA sequences, piRNAdb v1.7.6 [34] for piRNAs and miRBase Release 22.1 [33] for microRNAs. These databases were accessed in September 2021 and are most frequently used in sRNA studies, and specifically, they were used in the analyzed case studies.

For *Rattus norvegicus*, *Drosophila melanogaster* and *Mus musculus* piRNA and microRNA annotations, an additional UCSC liftOver procedure [36] was followed to map loci to the most recent genome version (from rn6 to rn7, from dm3 to dm6 and from mm10 to mm39, respectively).

Sequences in fasta format from the reference genome (getfasta sequences) that correspond to the annotation bed-file were obtained by bedtools getfasta version 2.27.1 [43]. Fasta sequences were mapped on the reference genome of the current version by hisat2 [44] with parameters *–no-spliced-alignment –no-softclip –mp 100000,100000 –rfg 100000,100000*, with a nearly 100% overall alignment rate. Alignment sam files were created and transformed into a tsv format table (alignment table), which included information on sequence ID, chr and start position of alignment, sequence, length of the sequence and number of sequence mappings to the reference genome.

Figure 6 summarizes all stages of data processing undertaken for annotation filtering.

The comparative loci table was constructed based on annotation file information. In the first step, columns with sRNA type, ID, locus, chromosome, start position, end position, strand, length (end–start position) were created. The second step consisted of the addition of getfasta sequences to their locus and creation of their reverse complement sequence. The most important, the third step was adding a fasta sequence to the locus with the same ID and checking if the getfasta sequence in forward or reverse orientation was equal to the fasta sequence. This step would determine the quality and further means of these emerging records of our final database. Then, information about the number of mappings of the sequence on the reference genome and whether the fasta sequence was mapped to the locus was added from the alignment table.

The final consensus database was built from the comparative loci table and alignment table. Information for consensus data was obtained using different approaches, depending on a particular fasta sequence and locus matching situation.

If the fasta sequence was equal to the getfasta sequence (*matched fasta&locus* case), then the locus from the annotation was accepted as the consensus locus, and the fasta sequence was accepted as the consensus sequence.

If the record had no fasta sequence for the ID from the annotation file (*locus only* case), the orientation-specified getfasta sequence was accepted as the consensus sequence.

If the record had only a fasta sequence that was mapped on the reference genome (*fasta only* case), the mapping position in the genome was accepted as the consensus locus.

If information on both locus and fasta was present and the getfasta sequence was equal to the part of the fasta sequence (*fasta & extended locus* case), the annotation locus was extended according to the fasta sequence and then accepted as the consensus locus, and the fasta sequence was accepted as the consensus sequence. The large number of records from this source may be related to liftOver transformation of the annotation file.

In case the fasta sequence was equal to the part of the getfasta sequence (*extended fasta & locus* case), the getfasta sequence, in turn, was accepted as a consensus sequence, and the locus from the annotation was accepted as a consensus locus.

When the part of the fasta sequence was equal to the part of the getfasta sequence (*extended fasta & extended locus* case), the annotation locus was extended according to the fasta sequence and then accepted as the consensus locus, and the union of the fasta and the getfasta sequences was accepted as a consensus sequence.

### 3.2. rRNA Processing

For all species, we employed information about sequences and annotation from the UCSC database for the current version of the reference genome. Data from Silva v.138 (accessed in September 2021) [45] were used as well; however, all obtained data were found in the UCSC database, so the Silva database was dismissed.

Fasta sequences from UCSC were mapped on the reference genome of the current version by hisat2 with parameters –no-spliced-alignment –no-softclip –mp 100000,100000 –rfg 100000,100000. The next steps were in the same manner as for sRNA types, described previously in Section 3.1, except that fasta sequences were added to the loci with the same ID and loci mentioned in the fasta file. The final consensus database was built from the comparative loci table and alignment table. Information for consensus data was obtained from only two sources.

If the fasta sequence was equal to the getfasta sequence (*matched fasta&locus* case), then the locus from the annotation was accepted as the consensus locus, and the fasta sequence was accepted as the consensus sequence.

If the record had only a fasta sequence that was mapped on the reference genome (*fasta only* case), the mapping position in the genome was accepted as the consensus locus.

### 3.3. tsRNA Processing

For *Homo sapiens*, we employed information from MINT database v2.0 and the hg19 reference genome (accessed in September 2021) [38]. For other species (*Mus musculus*, *Drosophila melanogaster*, *Caenorhabditis elegans*)) data from tRFdb (accessed in September 2021) [37] were recruited. Files with tRNA fragment sequences and annotation in tsv format for MINTbase and files in xls format from tRFdb were obtained and then files with sequences in fasta format (fasta sequences) and annotation files in bed format were created for each species. Genome coordinates were translated to the most recent genome version (from mm9 to mm39 for *Mus musculus*, from dm3 to dm6 for *Drosophila melanogaster*, from ce6 to ce11 for *Caenorhabditis elegans*, from hg19 to hg38 for *Homo sapiens*) with UCSC LiftOver.

Fasta sequences from MINTbase were mapped on the reference genome of the current hg38 version by hisat2 with parameters –no-spliced-alignment –no-softclip –mp 100000,100000 –rfg 100000,100000, with a 65.3% overall alignment rate. Next steps were processed in the same manner as for sRNA, described previously in Section 3.1.

Figure 7 summarizes all the stages of data processing undertaken for tsRNA annotations filtering.

The final consensus database was built from the comparative loci table and alignment table using the same manner to determine consensus data as described previously in Section 3.1.

### 3.4. Database Integration

At this point, transcripts with conflicting data due to in-database and inter-database (mature microRNA, piRNA, rRNA, tRNA) loci intersections were detected with the help of bedtools intersect (*Inbase conflict* and *Interbase conflict* correction events in Table 2). Conflicting data due to in-database intersections were removed from all datasets; conflicting data due to inter-database intersections were removed from annotations on mature microRNA, piRNA, rRNA, tRNA types.

Finally gtf-format files for all sRNA types and, separately, for mature microRNA, piRNA, rRNA, tRNA and tsRNA for five species were created. They contain information about ID, sequence and loci for each transcript.

### 3.5. Case Studies— sRNA

Small RNA sequence data of various sample sizes (n = 23, Donkin et al., 2016 [40], n = 24, Ingerslev et al., 2018 [41], n = 87, Hua et al., 2019 [42]) were obtained from materials associated with published studies [40,41,42], all accessed in September 2021. All reads were trimmed by cutadapt [46] following lab kit instructions. Then, reads with a length of 15–45 nt were mapped on the UniVec database by Hisat2 [44].

Then, trimmed reads were processed by SPORTS (accessed in September 2021) [19] with its default sRNA databases, and default settings for alignment were used. Moreover, reads were aligned to the reference genome by the bowtie aligner [30] (-v 0 -m 100 -k 1 –best –strata) with the following Rsubread [31] analysis (with gtf file using ITAS) to obtain counts for small RNA for further pipelines.

Differential expression analysis was performed by the DESeq2 (accessed in September 2021) [27] package with factors investigated in each article. Heatmaps and boxplots were built for small RNA for genome-alignment-based analysis (bowtie + Rsubread) with ITAS and for SPORTS default analysis (Appendix A).

### 3.6. Case Studies— Fragments

Counts for tRNA-derived items were obtained from the SPORTS pipeline directly with counts of small RNA. Rsubread was used to obtain reads, which were mapped on mature tRNA loci. These reads were processed with Kallisto (accessed in September 2021) [32] with k-mer length 11 with tsRNA from ITAS and then differential analysis was carried out with DESeq2. Heatmaps and boxplots were built for tsRNA for genome-alignment-based analysis (bowtie + Rsubread + Kallisto) with ITAS and for SPORTS default analysis (Appendix A).

### 3.7. Program Code Availability

All scripts used for the ITAS processing and analysis are available on Github (accessed on 25 April 2022), as well as the manual (accessed on 25 April 2022).

## 4. Discussion

Establishing best practices and analytic pipelines is important for sRNA expression analysis. In this study, we integrated data from separate databases on different sRNA types into ITAS transcript annotation in a commonly used gtf format for five species. The use of ITAS allows us to employ alignment- and pseudo-alignment-based bioinformatic approaches for transcriptomics analysis.

The conducted case studies using human sperm RNA-seq data [40,41,42] demonstrated the advantages of ITAS. Mapping of reads to ITAS, which unifies in a single gtf format database transcripts from all source databases, allowed for the identification of more significant transcripts as compared with the ’map and remove’ approach. In particular, for sRNA expression analysis, all cases revealed that the ITAS-based genome alignment approach identified more significant transcripts than the ’map and remove’ pipeline with default databases (11 vs. 66 for Donkin et al., 43 vs. 212 for Ingerslev et al., 26 vs. 242 for Hua et al.).

For tsRNA expression analysis, the ’map and remove’ pipeline was not able to identify any tsRNA for Hua et al.’s data, unlike the ITAS-based genome alignment pipeline, which identified 12 differentially expressed transcripts. Using Ingerslev et al.’s data, the ITAS-based genome alignment approach identified more significant transcripts compared to the ’map and remove’ pipeline (24 vs. 12 transcripts, respectively). However, using Donkin et al.’s data, the ’map and remove’ pipeline identified more significant transcripts compared to the ITAS-based genome alignment approach (5 vs. 3 transcripts, respectively). Results of case studies illustrate the importance of high flexibility in the choice of bioinformatic approaches to obtain a higher overall signal, which is exemplified through ITAS.

Moreover, we show the importance of the transcripts’ pre-filtering, which prevents the reporting of transcripts that conflict with other entries. For example, Leu-CAG tRNA was identified in the SPORTS-based ’map and remove’ analysis of Hua et al.’s data. This transcript was filtered out while forming ITAS as intersecting with the hsa-piR-32972 piRNA transcript. The filtered-out transcript cannot belong unambiguously to either Leu-CAG tRNA or hsa-piR-32972 piRNA.

There were several issues described above while constructing the ITAS. The main aim was to save as many transcripts as possible and to complete them with missing information. It is not only the strength but may in some cases also be the limitation of the ITAS. Sometimes, in-transcript conflicts can be caused by the LiftOver transformation of coordinates. The large number of transcripts with loci shorter than the fasta sequence only by one nucleotide may indicate a database-independent issue, which was successfully solved by loci extending. However, a greater discrepancy between loci and fasta may indicate the imperfection of existing databases or necessity for transcript rechecking and verification due to the information loss on loci following LiftOver transformation.

This is especially important for MINTbase (Human tsRNA), since there were no records with equal fasta sequence and locus lengths (Figure 3C). Moreover, 8120 tsRNA lost their loci after LiftOver and were added to ITAS by mapping on the reference hg38 genome; 1718 records had severe in-transcript conflicts and were filtered out (Figure 3D).

The extension of a locus or sequence leads to the formation of a new transcript that is considered correct. However, it is not presented in the existing databases or differs from known transcripts. Therefore, the validity of such transcripts is questionable.

The ITAS was checked for in-base and inter-base intersections, but not for intersections with transcriptomes. Thus, some new or now existing loci may be in conflict with other genes. In future, it should be checked further, and more data from newly published databases and existing annotations should be added to ITAS, such as data for other species or other small RNA types, including rsRNA, and data from the new database (MirGeneDB 2.1) [47].

## 5. Conclusions

We have identified several issues during the inference of existing databases containing information on sRNA transcript sequences and annotation for several species (*Homo sapiens*, *Mus musculus*, *Rattus norvegicus*, *Drosophila melanogaster*, *Caenorhabditis elegans*). Some transcripts had missing information on their sequences or loci; for others, their genome locus-retrieved sequence and database provided-sequence were not matching. Transcripts had both in-database and inter-database intersecting loci with other entries. This can pose problems towards the flexibility and robustness of transcriptomics analysis recruiting this annotation information.

To address these drawbacks, we established ITAS, a filtered, corrected and integrated database for five species, in the widely used gtf format.

## Figures and Tables

**Figure 1 ncrna-08-00030-f001:**
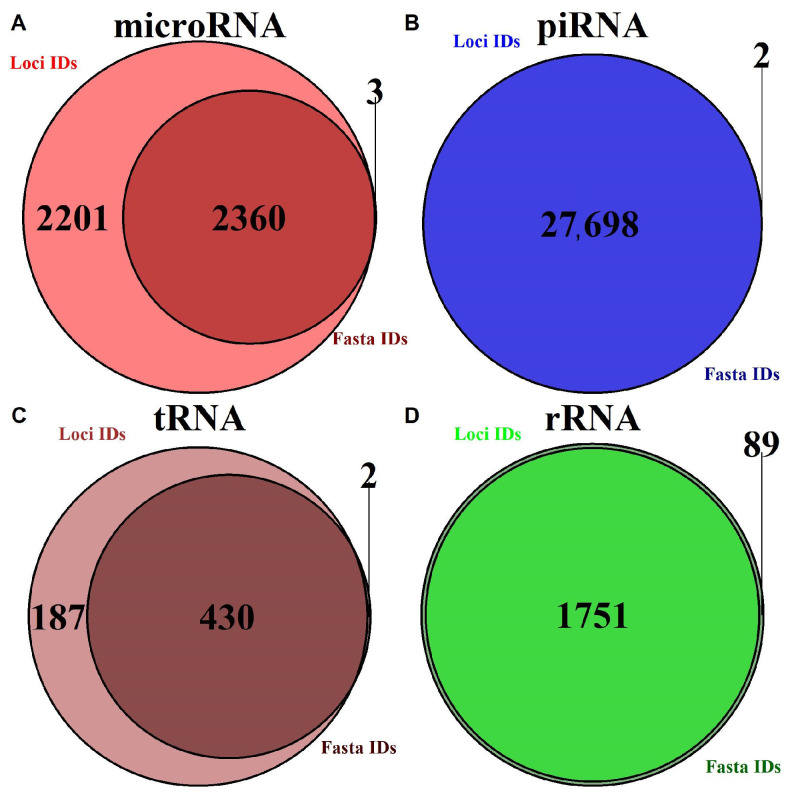
Visualization database completeness of both sequence data and loci data for various small RNA types. (**A**) For microRNA. (**B**) For piRNA. (**C**) For mature tRNA. (**D**) For rRNA.

**Figure 2 ncrna-08-00030-f002:**
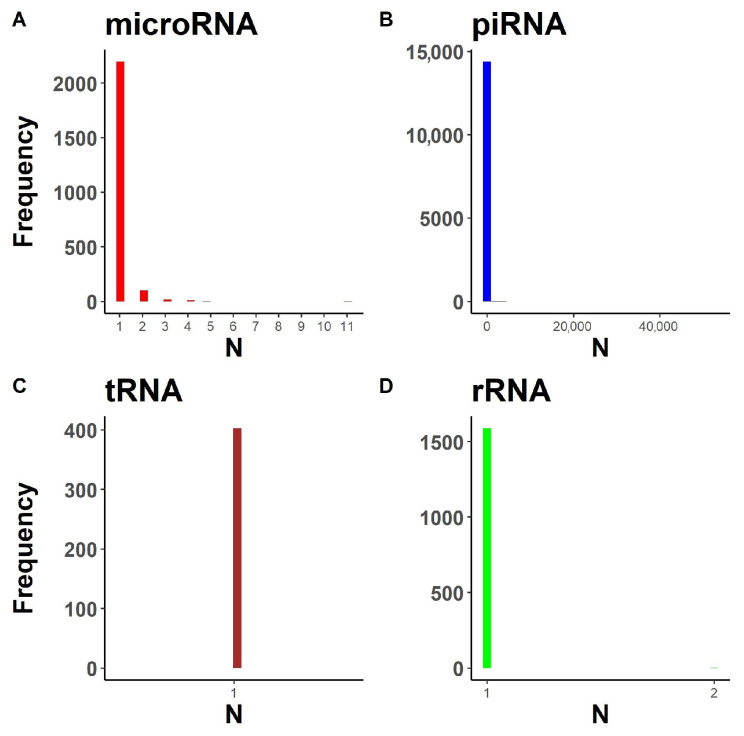
Distribution of loci number per transcript in the databases for various small RNA types. (**A**) For microRNA. (**B**) For piRNA. (**C**) For mature tRNA. (**D**) For rRNA.

**Figure 3 ncrna-08-00030-f003:**
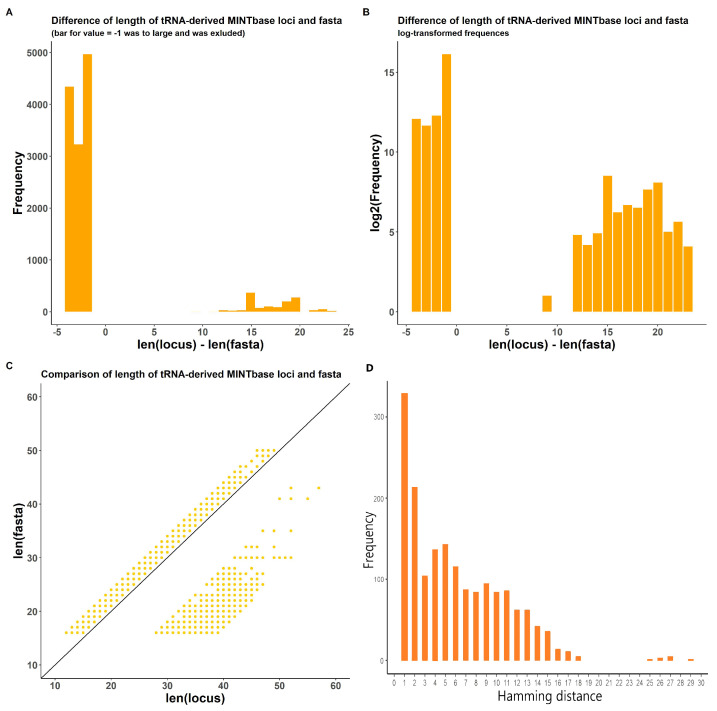
Statistics for database-delivered sequence (*fasta*) and genome locus-delivered sequence (*getfasta*) for MintBase transcripts. (**A**) Difference in lengths between fasta and getfasta in absolute values of frequency. Bar for value = −1 was larger than others and was excluded for visualization. (**B**) Difference in lengths between fasta and getfasta in log-transformed values of frequency. (**C**) Plot for lengths of fasta and getfasta for transcripts. (**D**) Hamming distance distribution for transcripts where fasta and getfasta sequences have mismatches.

**Figure 4 ncrna-08-00030-f004:**
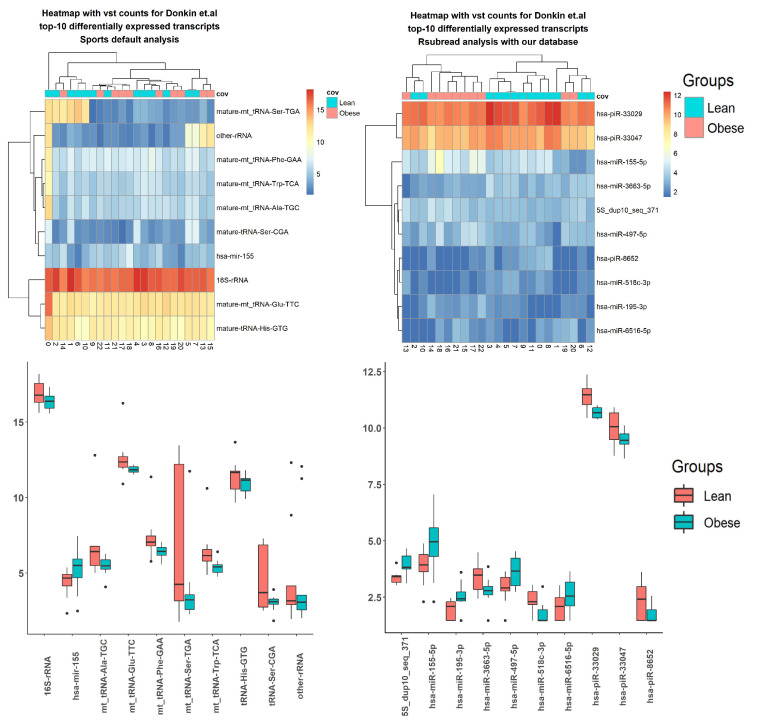
Top 10 differentially expressed small RNA (sRNA) using Donkin et al.’s data, processed with SPORTS vs. Integrated Transcript Annotation for Small RNA (ITAS)-based genome alignment (bowtie + Rsubread) pipelines.

**Figure 5 ncrna-08-00030-f005:**
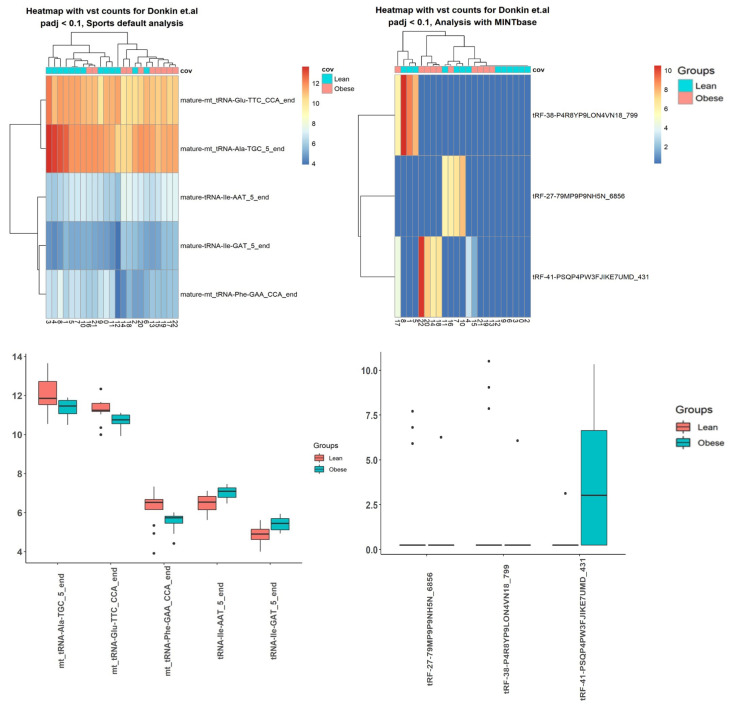
Top differentially expressed tRNA-derived small RNA (tsRNA) using Donkin et al.’s data, processed by SPORTS vs. ITAS-based genome alignment (bowtie + Rsubread + Kallisto) pipelines.

**Figure 6 ncrna-08-00030-f006:**
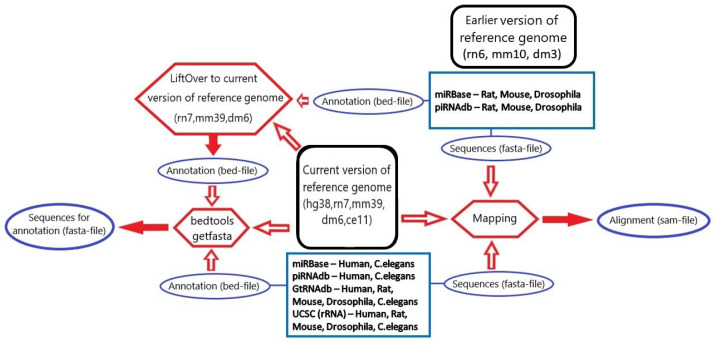
An overview of all steps for processing considered databases of small RNAs for 5 species.

**Figure 7 ncrna-08-00030-f007:**
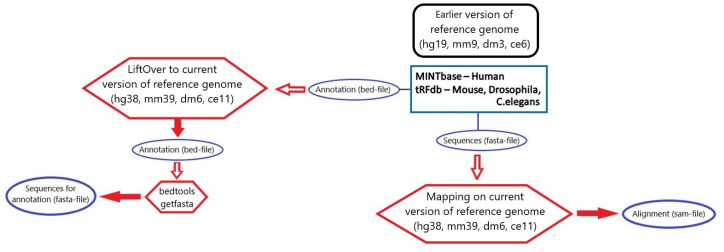
An overview of all processing steps for integration of tsRNA data for four species.

**Table 1 ncrna-08-00030-t001:** Statistics on completeness of human small RNA transcript entries in the databases prior to any correcting or filtering.

RNA Type	Database	Loci and Sequence	Loci Only	Sequence Only	Genome Version
precursor microRNA	miRBase	1002	913	2	hg38
mature microRNA	miRBase	1477	1405	1	hg38
piRNA	piRNAdb	812,343	0	2	hg38
tRNA	GtRNAdb	430	187	2	hg38
rRNA	UCSC	1752	0	186	hg38
tsRNA	MINTbase	125,285	0	0	hg19

**Table 2 ncrna-08-00030-t002:** Statistics on correction events in human RNA transcript entries: cases when both locus and sequence were present (*Loci & Seq*, no correction), only locus or only sequence (*Loci only*, *Seq only*, sequence or locus retrieved from genome); cases that required extending entry’s locus, sequence or both (*Ext Loci & Seq*, *Loci & Ext Seq*, *Ext Loci & Ext Seq*); cases with transcript loci intersections within same database (*Inbase conflicts*) and intersections between different databases (*Interbase conflicts*). * No intersection events were considered for tsRNAs, as fragments of the same tRNA naturally have intersecting loci.

RNA Type	Database	Loci & Seq	Loci Only	Seq Only	Ext Loci & Seq	Loci & Ext Seq	Ext Loci & Ext Seq	Inbase Conflict	Interbase Conflict
mature	miRBase	1291	1227	1	0	0	0	140	21
microRNA									
piRNA	piRNAdb	422,017	0	1	0	0	0	388,826	1268
tRNA	GtRNAdb	225	176	2	0	0	0	0	214
rRNA	UCSC	1591	0	186	0	0	0	37	126
tRNA-derived	MINTbase	0	0	8120	115,040	33	8	*	*

**Table 3 ncrna-08-00030-t003:** Statistics on correction events in tsRNA entries for various species: cases when both locus and sequence were present (*Loci & Seq*, no correction), only locus or only sequence (*Loci only*, *Seq only*), and cases that required extending entry’s locus, sequence or both (*Ext Loci & Seq*, *Loci & Ext Seq*, *Ext Loci & Ext Seq*).

Species	Database	Loci & Seq	Loci Only	Seq Only	Ext Loci & Seq	Loci & Ext Seq	Ext Loci & Ext Seq
*Homo sapiens*	MINTBase	0	0	8120	115,040	33	8
*Mus musculus*	tRFdb	335	0	0	0	0	8
*Drosophila melanogaster*	tRFdb	147	0	0	0	0	0
*Caenorhabditis elegans*	tRFdb	247	0	0	0	0	0

**Table 4 ncrna-08-00030-t004:** Statistics for unique transcript IDs for Integrated Transcript Annotation for Small RNA (ITAS), correction and filtration of intersection inside databases and between databases.

Mature						
Species	Genome Version	microRNA	piRNA	tRNA	rRNA	tsRNA
*Homo sapiens*	hg38	2330	14,439	403	1776	18,948
*Mus musculus*	mm39	1870	9715	1044	1376	13,105
*Rattus norvegicus*	rn6	616	7976	966	239	9797
*Caenorhabditis elegans*	ce11	397	8376	633	5	9411
*Drosophila melanogaster*	dm6	435	8296	154	93	8978

**Table 5 ncrna-08-00030-t005:** Statistics for unique transcript IDs for ITAS, after filtration and correction. Intersections between sRNA types were not filtered.

Mature						
Species	Genome Version	microRNA	piRNA	tRNA	rRNA	tsRNA
*Homo sapiens*	hg38	2543	14,605	619	1840	26,731
*Mus musculus*	mm39	1870	9739	1135	1430	65
*Rattus norvegicus*	rn6	647	8079	1173	274	-
*Caenorhabditis elegans*	ce11	424	8654	721	5	18
*Drosophila melanogaster*	dm6	481	500,536	295	165	22

**Table 6 ncrna-08-00030-t006:** Numbers of identified differentially expressed sRNA transcripts (sRNA) (*p*-value <0.05) and tRNA-derived small RNA (tsRNA) (adjusted *p*-value <0.1) for case studies processed with SPORTS pipeline with default annotation vs. genome alignment (bowtie + Rsubread/kallisto) pipeline based on ITAS.

Data	SPORTS, sRNA	ITAS, sRNA	SPORTS, tsRNA	ITAS, tsRNA
Donkin et al. [40]	11	66	5	3
Ingerslev et al. [41]	43	212	12	24
Hua et al. [42]	26	242	0	12

**Table 7 ncrna-08-00030-t007:** Top 10 differentially expressed small RNA (sRNA) transcripts and tRNA-derived small RNA (tsRNA) with adjusted *p*-value <0.1 for Donkin et al. case study processed with SPORTS pipeline with default annotation vs. genome alignment (bowtie + Rsubread/Kallisto) pipeline based on ITAS.

Transcript Name sRNA	SPORTS, *p*-Value
tRNA-Ser-CGA	3.23 $\times \;10^{-6}$
hsa-mir-155	0.001
other-rRNA	0.002
mt-tRNA-Glu-TTC	0.009
mt-tRNA-Phe-GAA	0.019
mt-tRNA-Trp-TCA	0.019
mt-tRNA-Ala-TGC	0.022
mt-tRNA-Ser-TGA	0.022
tRNA-His-GTG	0.028
16S-rRNA	0.035
**tsRNA**	**adjusted *p*-value**
tRNA-Ile-AAT-5-end	0.099
mt-tRNA-Glu-TTC-CCA-end	0.099
mt-tRNA-Ala-TGC-5-end	0.099
tRNA-Ile-GAT-5-end	0.099
mt-tRNA-Phe-GAA-CCA-end	0.099
**Transcript name** **sRNA**	**ITAS,** ***p*-value**
hsa-piR-33029	1.84 $\times \;10^{-5}$
hsa-miR-155-5p	0.002
5S-dup10-seq-371	0.003
hsa-miR-497-5p	0.006
hsa-piR-8652	0.006
hsa-miR-195-3p	0.007
hsa-piR-33047	0.008
hsa-miR-6516-5p	0.008
hsa-miR-3663-5p	0.012
hsa-miR-518c-3p	0.012
**tsRNA**	**adjusted *p*-value**
tRF-38-P4R8YP9LON4VN18-799	1.09 $\times \;10^{-13}$
tRF-27-79MP9P9NH5N-6856	1.29 $\times \;10^{-11}$
tRF-41-PSQP4PW3FJIKE7UMD-431	0.01

## Data Availability

Publicly available datasets were analyzed in this study. These data can be found here: GSE110190 (accessed on 25 April 2022), GSE74426 (accessed on 25 April 2022), GSE109475 (accessed on 25 April 2022). Integrated annotation gtf-format files for five species are available at the following link (accessed on 20 February 2022). All scripts used for constructing the ITAS and analysis are available at the following link (accessed on 20 February 2022).

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
