# Peer review of "ITAS: Integrated Transcript Annotation for Small RNA"

_ncrna, 2022, doi:10.3390/ncrna8030030_

Round 1
Reviewer 1 Report
The authors presented the integrated transcript annotation for small RNA (ITAS) attempting to resolve the data quality issues among exiting small RNA (sRNA) databases. In their manuscript, four different issues were noticed including 1) incomplete annotation, either loci or sequences, for certain transcripts and 2) in-transcript, 3) in-database transcript, and 4) inter-database transcript data conflicts. Collectively, the data in this study has the potential to facilitate transcriptomic analysis of sRNAs, and many non-coding RNA experts may find useful for their future research. A few recommendations listed below.
The data resource used to compile the information of miRNAs was outdated. miRBase had not been updated since 2019 and other newer databases are available, e.g., MirGeneDB 2.1 (2022) and RNA Atlas (2021). FANTOM5 by RIKEN had also provided an expression atlas of miRNAs. The authors should have performed an exhaustive and inclusive search for all sRNA biotypes before making claims of data incompletion.
Second, in your Table 1, all sequences of human precursor and mature miRNAs had been made available at miRBase FTP site (https://www.mirbase.org/ftp.shtml). The sequences of all tRNAs at GtRNAdb are also available at http://gtrnadb.ucsc.edu/genomes/eukaryota/Hsapi19/Hsapi19-seq.html. The rRNA sequences can be downloaded from UCSC Table Browser at ease. Please confirm whether this claimed data incompletion still holds.
Third, mature miRNAs of identical sequences can be derived from different precursor miRNAs located at a different genomic location including sex chromosomes. This should not be deemed as data conflicts but biology nature. The authors should confirm whether the claimed in-transcript data conflicts still hold for all sRNA biotypes after consideration of this fact. In addition, in Figure 3, please make sure these transcripts are not multi-exonic. After intron sequences are spliced out, it is not surprising that the database-derived sequences have different lengths as those genome locus-delivered sequences.
Author Response
Reviewer 1
Response:
We thank the reviewers for their thoughtful and constructive comments on our paper. We offer a point-by-point response to each comment below. The revised text of manuscript was highlighted in blue.
We have also followed the Reviewers’ request to improve English. Looking forward to your consideration.
Alexey Stupnikov and Oleg Sergeyev, on behalf of co-authors.
Comments and Suggestions for Authors
The authors presented the integrated transcript annotation for small RNA (ITAS) attempting to resolve the data quality issues among exiting small RNA (sRNA) databases. In their manuscript, four different issues were noticed including 1) incomplete annotation, either loci or sequences, for certain transcripts and 2) in-transcript, 3) in-database transcript, and 4) inter-database transcript data conflicts. Collectively, the data in this study has the potential to facilitate transcriptomic analysis of sRNAs, and many non-coding RNA experts may find useful for their future research. A few recommendations listed below.
- The data resource used to compile the information of miRNAs was outdated. miRBase had not been updated since 2019 and other newer databases are available, e.g., MirGeneDB 2.1 (2022) and RNA Atlas (2021). FANTOM5 by RIKEN had also provided an expression atlas of miRNAs. The authors should have performed an exhaustive and inclusive search for all sRNA biotypes before making claims of data incompletion.
Response:
We thank the reviewer for the suggestion. We agree that miRBase has not been updated since 2019. Nevertheless, it contains a large number of miRNA sequences and coordinates of recent genome version (1917 precursors and 2883 mature miRNA) and this database is most frequently used in snRNA studies, including those presenting data for the case-studies we investigated in our paper (Donkin et al, 2016, Ingerslev et al., 2018 and Hua et al., 2019).
Recently published in January 2022 (https://doi.org/10.1093/nar/gkab1101) MirGeneDB 2.1 database contains RNA sequences and loci of precursor and mature RNA for all species included in our paper. For Human, it consists of 567 and 630 RNA sequences and coordinates of precursor and mature RNA respectively. However, this recent database was not available for us when we were preparing the manuscript.
RNA Atlas consists of Human 4105 and 5465 RNA loci for 4074 precursor and 5280 mature RNA unique IDs, respectively. Fantom5 consists of Human 2697 and 2700 RNA loci for precursor and mature RNA, respectively, for hg19, not for hg38. It also has data for the Mouse. However, both RNA Atlas and Fantom 5, do not contain transcript sequences.
Thus, we chose MiRBase for ITAS as the miRNA database with a large number of transcripts and loci coordinates using a recent reference genome (for Human), and the most complete data among other databases. We added clarification of the reasons why we chose databases that were used, in the Methods (lines 167-169):
“Files with mature transcripts sequences in fasta format (referred to as fasta sequences) and annotation of transcripts in bed-format (loci) were obtained from corresponding databases: GtRNAdb Data Release 19 (June 2021) {Chan et al, 2016} for mature tRNA sequences, piRNAdb v1.7.6 {Piuco et al, 2021} for piRNAs and miRBase Release 22.1 {Kozomara et al, 2014} for microRNAs. These databases were accessed in September 2021 and and most frequently used in sRNA studies, and specifically, they were used in the analyzed case studies.”. (lines 164-169).
Additionally, we plan to use recent databases, including MirGeneDB 2.1, for updates of the current version of ITAS in our next releases:
“In future it should be checked further as well as more data from newly published databases and existing annotations should be added to ITAS, such as data for other species, other small RNA types, including rsRNA and data from the new database (MirGeneDB 2.1)” (lines 334-336).
- Second, in your Table 1, all sequences of human precursor and mature miRNAs had been made available at miRBase FTP site (https://www.mirbase.org/ftp.shtml). The sequences of all tRNAs at GtRNAdb are also available at http://gtrnadb.ucsc.edu/genomes/eukaryota/Hsapi19/Hsapi19-seq.html. The rRNA sequences can be downloaded from UCSC Table Browser at ease. Please confirm whether this claimed data incompletion still holds.
Response: We confirm we used the databases the Reviewer suggested. We also added the exact access date to the text (lines 67, 167-169) :
“Several data resources have been selected as sources of annotations for specific RNA biotypes. They include miRBase for microRNA, piRNAdb for piRNA, GtRNAdb for tRNA, UCSC database for rRNA, tRFdb for Mus musculus, Drosophila melanogaster, and Caenorhabditis elegans tsRNA, and MINTbase for Homo sapiens tsRNA. All mentioned databases were accessed in September 2021.”(Lines 63-67).
“Files with mature transcripts sequences in fasta format (referred to as fasta sequences) and annotation of transcripts in bed-format (loci) were obtained from corresponding databases: GtRNAdb Data Release 19 (June 2021) {Chan et al, 2016} for mature tRNA sequences, piRNAdb v1.7.6 {Piuco et al, 2021} for piRNAs and miRBase Release 22.1 {Kozomara et al, 2014} for microRNAs. These databases were accessed in September 2021 and and most frequently used in sRNA studies, and specifically, they were used in the analyzed case studies”. (lines 164-169)
- Third, mature miRNAs of identical sequences can be derived from different precursor miRNAs located at a different genomic location including sex chromosomes. This should not be deemed as data conflicts but biology nature. The authors should confirm whether the claimed in-transcript data conflicts still hold for all sRNA biotypes after consideration of this fact.
Response: We agree with the reviewer that mature miRNAs of identical sequences can be derived from different precursor miRNAs located at a different genomic locations. Indeed, precursor and mature transcripts may cause formal conflict in a database, being different versions of the same biological object. However, we took this circumstance into consideration and extracted only mature miRNA for our database. We clarified this fact by explicitly stating the miRNA type (mature miRNA) in Tables 2,4,5.
- In addition, in Figure 3, please make sure these transcripts are not multi-exonic. After intron sequences are spliced out, it is not surprising that the database-derived sequences have different lengths as those genome locus-delivered sequences.
Response: We appreciate this suggestion. We agree that the length of database-derived sequences and genome locus-delivered sequences could be different due to splicing. However, in our study, we identified the main reasons for the observed length differences. The first cause was the ambiguity in the format of loci coordinates in the initial database (which can be loose up to several bases), and the second cause was the translation process of coordinates from one genome version to another. We have added a comment on these effects to the manuscript (lines 86-89):
“Another identified problem is that for many transcripts (tsRNAs in MINTbase for Homo sapiens and various small RNA types in other species) the length of the locus does not match the length of the corresponding fasta sequence (Figure 3 A,B,C). This occurs due to the lack of precision in initial database loci coordinates and LiftOver transformation for genomic coordinates from hg19 to hg38 genome versions. These effects were corrected in the final ITAS version. In addition to this, 1718 transcripts in MINTbase had significant differences between sequences in database-derived fasta file and actual nucleotide sequences in corresponding genome loci (Figure 3 D). (Lines 84-91)
Thus, per Revewer's suggestions, we have revised the Results, Methods, and Discission sections and clarified in the section of Materials and Method that all code, scripts, and manual for ITAS are located on the Github. Additionally, we once again carefully checked the article with the help of a co-author who is a native English speaker.
Reviewer 2 Report
In this manuscript, the authors introduce an integrated way to annotate small RNAs. The authors comprehensively cover different types of small RNAs by collecting currently scattered data regarding each type of small RNAs into one data format, Integrated transcript annotation for small RNA (ITAS). The methods the authors employed are comprehensive, but this tool is not available to public. More specific comments are listed below:
Major points:
[1] “2.3. Case studies”. This is the most important section of the manuscript. All the figures and tables should be moved to the main text and move most of the data from the previous parts as supplementary data.
[2] The authors must disclose the computational code, programs, and pipeline used in this study in a public domain, such as GitHub.
[3] The authors should make Integrated transcript annotation for small RNA (ITAS) as computer program so that the users can easily use this tool.
Minor points:
(1) The data access date for each data set is missing.
Author Response
Reviewer 2
We thank the reviewers for their thoughtful and constructive comments on our paper. We offer a point-by-point response to each comment below. The revised text of manuscript was highlighted in blue.
We have also followed the Reviewers’ request to improve English. Looking forward to your consideration.
Alexey Stupnikov and Oleg Sergeyev, on behalf of co-authors.
Comments and Suggestions for Authors
In this manuscript, the authors introduce an integrated way to annotate small RNAs. The authors comprehensively cover different types of small RNAs by collecting currently scattered data regarding each type of small RNAs into one data format, Integrated transcript annotation for small RNA (ITAS). The methods the authors employed are comprehensive, but this tool is not available to public. More specific comments are listed below:
Major points:
[1] “2.3. Case studies”. This is the most important section of the manuscript. All the figures and tables should be moved to the main text and move most of the data from the previous parts as supplementary data.
Response: We agree the case study is indeed the most important section of a paper, and the submitted version lacked visual and supporting materials. Our initial concern was that the number of figures and the size of the tables for all the case studies would make the manuscript overloaded in this section.
To improve the manuscript in a way the reviewer suggested we included the materials for one case study (Figures 4,5 and Table 7) to illustrate the claims we make in the results section (lines 150-154, pages 7-9). Materials for other case studies (Suppl. Figures 1-6 and Suppl. Tables 5-15) remain in the supplementary since they do not suggest grounds for additional claims, although reinforce existing ones.
The supporting materials for other sections were left in the text, as we believe they provide vital aid to our exploratory analysis, illustrate the key problems of existing databases, and expose the necessity for improvements of existing databases suggested in the paper.
[2] The authors must disclose the computational code, programs, and pipeline used in this study in a public domain, such as GitHub.
Response: the reviewer raises an important point of analysis reproducibility. We agree that the methods and scripts used for the computational analysis should be made available to ensure the study and results are reproducible.
For this reason, we initially included the link to Github repository in the ‘Data availability statement’ section containing all relevant scripts and code. We agree with the reviewer that this fact needs to be stressed in the main body of the manuscript as well. To address this issue, we clarified that all code and scripts are located on the Github in the section of Materials and Methods, in addition to directly listing links in the Data Availability Statement section:
“All scripts used for the ITAS processing and analysis are available on Github {https://github.com/EpiEpiMSU/ITAS_scripts} as well as manual {https://github.com/EpiEpiMSU/ITAS#readme }” (lines 285-287).
[3] The authors should make Integrated transcript annotation for small RNA (ITAS) as computer program so that the users can easily use this tool.
Response: We thank the reviewer for this comment. We agree that facilities for bioinformatics analysis, whether it is a tool or a data resource, need to be well set and documented to facilitate their usage and access for the scientific community. As we stated in the manuscript, the generated transcripts annotation is a data resource that, although not being itself a computational method, can be employed by existing bioinformatics programs for Differential Expression Analysis. For this reason, gtf format for the annotation file was chosen, as this format is used most frequently in transcriptomic pipelines.
To address the Reviewer's suggestion, we added a manual to the annotation, illustrating its processing and usage in several examples (https://github.com/EpiEpiMSU/ITAS/blob/main/README.md)
Minor points:
(1) The data access date for each data set is missing.
Response: We have added the information about access date: (lines 167-169, 217-218, 234, 235, 265, 268, 273, 280):
“accessed in September 2021”
Thus, per Revewer's suggestions, we have revised the Results, Methods, and Discission sections and clarified in the section of Materials and Method that all code, scripts, and manual for ITAS are located on the Github. Additionally, we once again carefully checked the article with the help of a co-author who is a native English speaker.
Round 2
Reviewer 1 Report
The authors have adequately addressed all points raised in my previous report, and I am satisfied with their responses.
Reviewer 2 Report
I have no further comment to make.